# surveydown: An open-source, markdown-based platform for programmable and reproducible surveys

**Pingfan Hu**, **Bogdan Bunea**, **John Paul Helveston***

Department of Engineering Management and Systems Engineering, George Washington University, Washington, District of Columbia, United States of America

\* jph@gwu.edu

**Data availability statement:** The source code for the package described in this paper can be found at https://github.com/surveydown-dev/surveydown/.

## Abstract

This paper introduces the surveydown survey platform. With surveydown, researchers can create surveys that are programmable and reproducible using markdown and R code, leveraging the Quarto publication system and R Shiny web framework. While most survey platforms rely on graphical interfaces or spreadsheets to define survey content, surveydown uses plain text, enabling version control and collaboration via tools like GitHub. The package renders surveys as interactive Shiny web applications, allowing for complex features like conditional skip logic, dynamic question display, and complex randomization. The package supports a diverse set of question types and formatting options and users can leverage Shiny's powerful reactive programming model to create a wide variety of interactive features. As an open-source platform, surveydown provides researchers full control over their survey implementation, including the survey application as well as where and how the resulting response data are stored. Workflows are entirely reproducible and integrate seamlessly with existing workflows for data collection and analysis in R.

## Introduction

Survey research is integral to many fields, and researchers have a wide variety of software platforms to choose from depending on their needs. Those needs often extend well beyond the basic feature set of the survey software and include budgetary constraints (i.e. using a free or paid product), transparency (e.g., whether the platform is open-source), the user interface, the ability to collaborate across teams, the ability to control access to the raw data, and the learning curve associated with using the platform, among other considerations. These diverse requirements create a complex decision landscape for survey researchers seeking a software solution that meets their needs. Although there are many options to choose from, most impose fundamental limitations on reproducibility, collaboration, and integration with data analysis workflows. These limitations can impede scientific rigor, increase costs, and create barriers to effective research practices.

**Funding:** This work was partially supported by a grant from the Alfred P. Sloan Foundation (https://sloan.org/), Grant Number G-2023-20976 awarded to PI John Paul Helveston. The funders did not play a role in the study design, data collection and analysis, decision to publish, or preparation of the manuscript. There was no additional external funding received for this study.

**Competing interests:** The authors have declared that no competing interests exist.

Existing survey platforms typically rely on graphical user interfaces (GUIs) or spreadsheets (XMLForms) to define survey content, making version control, collaboration, and reproducibility difficult or impossible. Commercial platforms often require expensive licenses, placing them out of reach for many researchers and students. Additionally, most platforms offer limited control over where and how response data is stored, raising concerns about data ownership and long-term accessibility. Finally, few platforms integrate seamlessly with modern data analysis workflows, often requiring manual data export and reformatting before analysis can begin.

This paper introduces surveydown, an open-source survey platform and software package for the R programming language [1] that addresses these limitations through several key innovations. First, surveydown employs a plain text, markdown-based system for defining surveys building on the Quarto publication system [2], enabling complete reproducibility and version control through tools like Git. Second, surveydown allows real-time code execution during survey administration by leveraging the R Shiny web framework [3], enabling complex and highly customized surveys with features such as dynamic question generation, complex randomization, and conditional logic that few existing platforms can match. Third, while most survey platforms employ an "all-in-one" design where a single application or website is used to design the survey, field it, and store the data, surveydown embraces a disaggregated design where researchers maintain complete control over their survey implementation and data storage.

The surveydown platform is particularly valuable for researchers who need reproducible survey designs, require complex survey functionality, or want to maintain complete control over their survey data, all without needing advanced web development skills, such as knowledge of JavaScript. The platform is accessible to users with basic R knowledge, while offering advanced capabilities for those with more experience in programming or the Shiny web framework. The approach aligns with modern reproducible research practices and integrates seamlessly with R-based data analysis workflows.

The remainder of this paper details the software design (Section 2), describes its key advantages and features compared to existing platforms (Section 3), and concludes with a discussion of community adoption, future directions, and limitations (Section 4).

## Software design

### Overall architecture

The surveydown project is both an R package and survey platform that leverages three open source technologies: Quarto for survey design, Shiny for the web framework, and PostgreSQL for data storage. The `surveydown` R package provides functions and the control logic to pull these technologies together into a cohesive survey platform. Available through the Comprehensive R Archive Network (CRAN), the `surveydown` package can be installed using `install.packages(``surveydown'')` in the R console and is available via a MIT license. Fig 1 below is an illustration of the core technologies that form the surveydown platform.

Every surveydown survey consists of a survey document and a web application, defined in two separate files named **survey.qmd** and **app.R**. These files must be in the same directory and have these precise file names as the surveydown package searches for them in the working directory. To make multiple surveys, users should organize each survey into a separate folder.

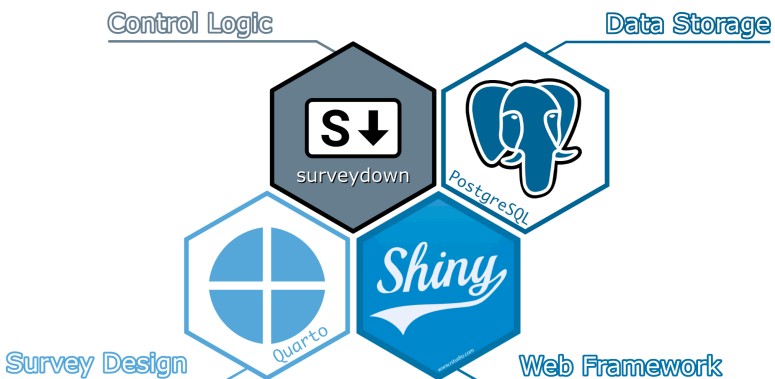

**Fig 1. Core technologies in the surveydown survey platform.** Quarto for survey design, Shiny for the web framework, and PostgreSQL for data storage. The `surveydown` R package ties them together into a cohesive survey platform.

The **survey.qmd** file is a standard Quarto document. Quarto is an open-source publishing system developed by Posit PBC that enables users to combine markdown-formatted text and code chunks into single documents (.qmd files) that can be rendered into a variety of different outputs, such as html pages, pdf documents, and even presentation slides and websites [2]. With surveydown, users define all of the main survey content using plain text (markdown and code chunks) in the **survey.qmd** file, including pages, text, images, questions, and navigation buttons.

The **app.R** is a standard R script defining a Shiny web application. The `shiny` R package allows users to build interactive web applications and dashboards using only R code, enabling users to create dynamic data visualizations and web-based tools without knowing web programming languages like JavaScript [3]. With surveydown, users define a Shiny application in the **app.R** file that includes global settings (libraries, database configuration, etc.) and server configuration options (e.g., conditional page skipping or question display).

The `surveydown` R package provides functions for defining survey content (e.g., survey questions, navigation buttons, etc.) as well as the overall server logic to drive the Shiny web application. Once a user is done defining the content in their survey, the Shiny application renders the **survey.qmd** Quarto document into a static html document, parses the document into survey pages, then serves each page in an interactive web application. The package also contains logic for controlling the storage of respondent data as it comes in once the survey is fielded. In the next section, we use an expositional example to showcase the construction of a minimum survey and provide a flow diagram to illustrate the overall logic flow of the surveydown platform for a typical survey.

## Expositional example

This section presents an expositional example of a two-page survey to explain the basic structure of the the **survey.qmd** and **app.R** files in a typical surveydown survey, followed by a flow diagram to explain the overall logic flows of what happens under the hood. The code below is an example **survey.qmd** file.

```
---
format: html
echo: false
warning: false
---

```{r}
library(surveydown)
```

::: {.sd_page id=welcome}

# Welcome to our survey!

```{r}
sd_question(
  type  = "mc",
  id    = "penguins",
  label = "What's your favorite penguin?",
  option = c(
    "Adélie"    = "adelie",
    "Chinstrap" = "chinstrap",
    "Gentoo"    = "gentoo"
  )
)

sd_next()
```

:::

::: {.sd_page id=end}

This is the last page of the survey.

```{r}
sd_close()
```

:::
```

At the top of the file is the YAML header, which defines several options to control the rendering process—namely, that the file should render into an html file (`format: html`), and that any code that is run in the file should not display the code itself or any warning messages when it runs (`echo: false` and `warning: false`). After loading the `surveydown` package, the rest of the file defines two pages: one with a multiple choice question, and another that is the ending page.

Pages are defined using three colon symbols `:::`, called a "fence", along with a `.sd_page` class definition and a page `id`. In the above example, the first page is defined as `::: {.sd_page id=welcome}`, where the `id` is set to `welcome`. In between this and the closing page `:::` symbol, users can insert content (e.g., text, images, links, etc.) using markdown formatting along with R code chunks to insert content defined using `surveydown` package functions.

Questions are defined using the `sd_question()` function. In the above example, the `type = ``mc"` argument is used to define a multiple choice question. The package supports a wide variety of question types, discussed in detail later in the paper. The `id` argument is set to `"penguins"`, which is the name that will be used to store the respondent data for this question. Finally, the `option` argument defines the multiple-choice options as a named vector, where the names are what respondents see and the values are what is stored in the data. Built-in question types include:

- `text`: text input type.
- `textarea`: textarea input type.
- `numeric`: numeric input type.
- `mc`: multiple choice type.
- `mc_buttons`: button version of `mc`.
- `mc_multiple`: multiple choice type with multiple selections.
- `mc_multiple_buttons`: button version of `mc_multiple`.
- `select`: drop down select type.
- `slider`: slider input type.
- `slider_numeric`: slider input type with numeric value, supporting single input or a range.
- `date`: date input type.
- `daterange`: daterange input type.
- `matrix`: matrix input type, containing a combination of `mc`. questions sharing a same set of options.

In addition to the question, the `sd_next()` is used inside the same code chunk to define a next button, which will by default navigate to the next page. Users can also provide an optional `next_page` argument to navigate to other survey pages if desired, using the page `id` as the `next_page` value. For now, surveydown only supports forward navigation as backwards navigation requires careful consideration of potential skipping logic that can create navigational loops, though adding support for a back button is on the development roadmap. Finally, the end page has a single sentence followed by the `sd_close()` function in another code chunk to insert a closing button that ends the survey. Fig 2 shows what the resulting two survey pages look like when rendered in a live survey app.

While the **survey.qmd** file defines the survey content, the **app.R** file renders the **survey.qmd** file into an interactive web application via the R `shiny` package. A minimal **app.R** file needs to contain code to 1) make the database connection to store respondent data, 2) define the user interface, 3) define the server, and 4) launch the app. The code below is an example of a minimal **app.R** file:

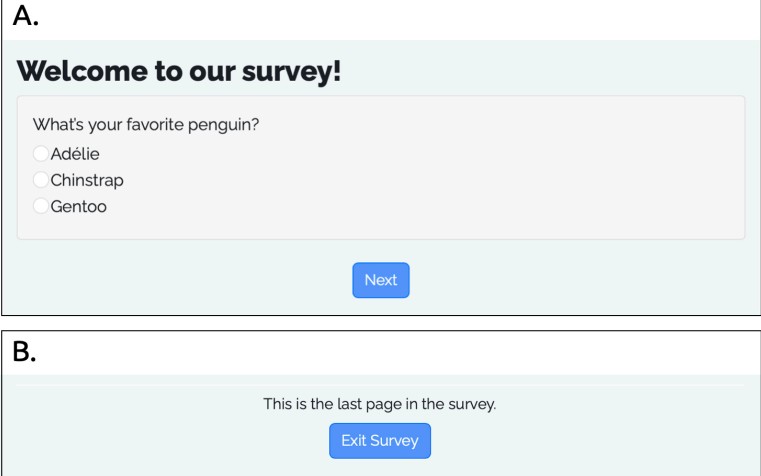

**Fig 2. Screenshots of the rendered survey pages in the above example survey.** A: The "Welcome to our survey!" text is in large, bold font because it is defined as a level 1 header, using the # symbol. The multiple choice question displayed is defined by the `sd_question()` function, and the "next" button is defined by the `sd_next()` function. B: The "Exit Survey" button is defined using the `sd_close()` function.

```r
library(surveydown)

# Database Credentials (Run in R Console)
# sd_db_config()

# Connect to Database
db <- sd_db_connect()

# Define the ui (processes the survey.qmd file)
ui <- sd_ui()

# Define the server
server <- function(input, output, session) {
  sd_server(db = db)
}

# Launch Survey
shiny::shinyApp(ui = ui, server = server)
```

After loading the `surveydown` package, the first few lines set up the database configuration. While any PostgreSQL database can be used for data storage, we recommend https://supabase.com as a free, open-source, cloud-based option. The `sd_db_config()` function can be run in the R console to store the database credentials in a local **.env** file, which include the host, port, database name, user name, password, and table name. Once the credentials are saved, the `sd_db_connect()` function is used to make a connection. In this example, the connection is created as the `db` object, which is then passed to the `sd_server()` function inside the server definition. Note that users should not store any of

the credentials in the **app.R** file; rather, once the **.env** file is created, it will be used to make the database connection.

After making the database connection, the user interface (`ui`) and `server` are defined, which are required components for any Shiny application. The `ui` is created with the `sd_ui()` function, which does two things. First, it renders the **survey.qmd** file and parses it into the components needed for the survey, which are stored in a local **_survey** folder. This function only re-renders the survey content if changes to the **survey.qmd** are detected or if required components are missing. Second, it sets up a placeholder user interface to display the rendered content, which is handled in the `server()` function.

The `server()` function takes the `input`, `output`, and `session` arguments, which are standard for any Shiny application. Inside, we call the `sd_server()` function, which is the primary `surveydown` function for controlling the survey logic, such as page navigation, data handling, etc. The `sd_server()` function has many optional arguments to fine-tune the control of the survey logic, and other code can also be included inside the `server()` function for other purposes, such as setting conditions for displaying specific questions or skipping forward to other pages in the survey. Some of these options are discussed in the section of Programmable interactivity via shiny.

The final line in the file calls the `shinyApp()` function, which is the standard `shiny` package command to launch the Shiny application using the `ui` and `server` components. Users can run the application locally to test it for functionality. Once it is ready to be sent to respondents, the application can be deployed online using a variety of hosting services, such as shinyapps.io, Posit Connect Cloud, and Heroku. To deploy, using the `deployApp()` function from the `rsconnect` package:

```
rsconnect::deployApp(appName = "your_app_name")
```

Fig 3 below illustrates the overall logic flow of a typical survey using the surveydown platform, highlighting the three primary actions in the **app.R** file: connecting to a database with `sd_db_connect()`, rendering the **survey.qmd** file and creating the main UI container with `sd_ui()`, and serving the survey pages and updating the database with `sd_server()`. As the diagram illustrates, the survey designer only need to edit the **app.R** and **survey.qmd** files to define the survey content, while the `surveydown` package functions handle the survey web application implementation and database management.

While this example illustrates the basic structure of a surveydown survey, the platform offers extensive functionality beyond what is shown here, including conditional display logic, page skipping based on responses, randomization of content, custom interactive elements, and robust data management features. These more advanced features, which leverage the full power of R and the Shiny framework, enable researchers to create sophisticated survey instruments that can adapt to respondent inputs in real-time.

## Key advantages and comparison with alternatives

The surveydown platform offers several advantages over traditional survey platforms: it is composed entirely using free and open-source software, it enables a fully reproducibility survey design experience via markdown and R code, and it offers enhanced interactivity and extensive customization via Shiny. Furthermore, its disaggregated architecture allows the researcher control over where and how the survey application and data storage are hosted, providing fine-tuned control over the overall survey implementation.

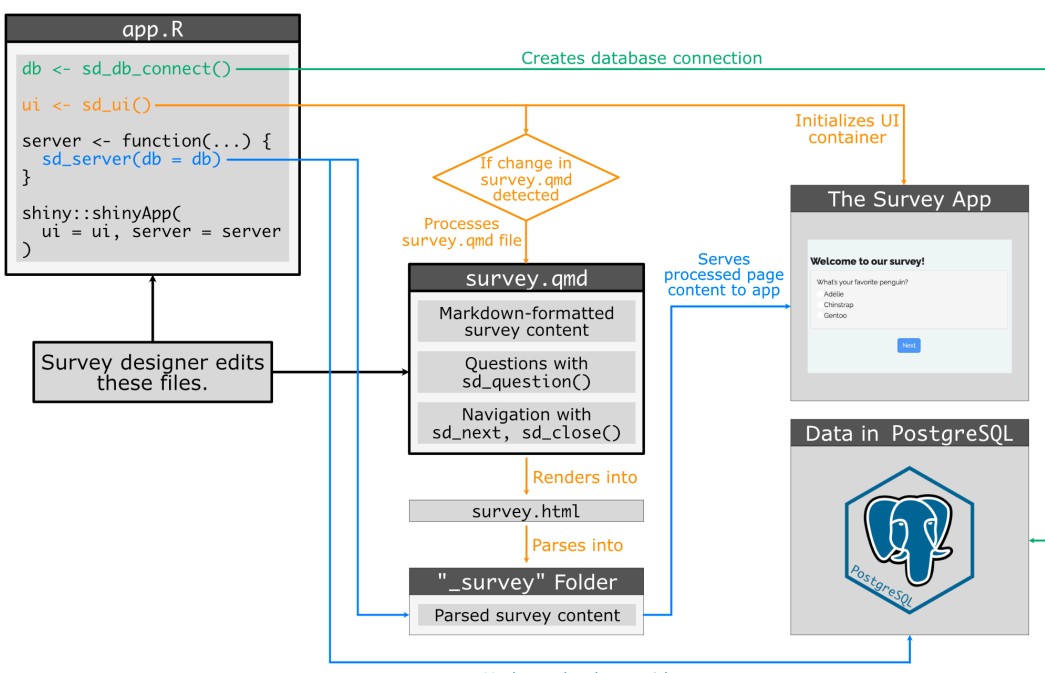

**Fig 3. Logic flow diagram of the surveydown survey platform.**

## Leveraging mature open-source technologies

The selection of R, Quarto, Shiny, and PostgreSQL as the foundational technology stack for surveydown was deliberate, considering the specific needs of survey researchers and the advantages these technologies provide over alternatives.

R [1] was chosen as the primary programming language for several key reasons. First, R has an established presence in the social sciences, where much of survey research takes place. Many researchers in disciplines such as sociology, psychology, political science, and economics are already trained in R for statistical analysis, reducing the learning curve for new users and facilitating integration with existing workflows. Second, R's broader ecosystem includes extensive packages for data manipulation, visualization, and analysis, making it ideal for a platform that aims to connect survey design directly to data analysis. For the case of surveydown, R has mature integration with connecting to PostgreSQL databases. Third, R's functional programming and lazy evaluation paradigm makes the language well-suited for controlling survey operations such as randomization, question generation, and conditional logic.

Quarto [2] was chosen as the framework for defining survey content in plain text files. As an evolution of R Markdown, Quarto combines the simplicity of markdown with deep integration of executable code. This combination is well-suited for survey contexts where textual content (instructions, questions, explanations) must be interspersed with functional components (questions, navigation, conditional elements). Quarto's ability to render content into static HTML serves as a critical intermediate step in the surveydown workflow, providing a consistent foundation for the dynamic Shiny application to build upon. Additionally, Quarto's widespread adoption in scientific publishing creates transferable skills as researchers already using Quarto for writing, presentations, or websites can apply that knowledge directly to

survey design in surveydown. Finally, surveydown users will directly benefit from all future innovations and improvements to Quarto over time with limited adaptations needed in the surveydown source code.

Finally, we chose Shiny [3] as the web framework for several reasons. First, it allows for the creation of interactive web applications without requiring knowledge of JavaScript, HTML, or CSS, making sophisticated survey functionality accessible to researchers without web development expertise. Second, with over a decade of development since its initial release in 2012, Shiny has matured into a robust framework with extensive documentation, community support, and a rich ecosystem of extensions. This maturity translates into reliability and sustainability for surveydown as a platform. Third, Shiny's reactive programming model is particularly well-suited to surveys, where changes in one part of the application (e.g., a respondent selecting an answer) can trigger updates elsewhere (e.g., displaying conditional questions or updating dynamic content).

The combination of these technologies creates a synergy that would be difficult to achieve with other technology stacks. Furthermore, by leveraging open-source technologies, the surveydown project also embraces open-source. Making the `surveydown` R package open-source not only allows researchers to inspect the underlying code to understand how their surveys functions, but also allows the community of users to contribute improvements, bug fixes, and new features. As of the composition of this paper, the `surveydown` R package has reached 118 GitHub Stars, 38 addressed issues, and 52 discussions led by users with 3,703 downloads from CRAN. In addition, contributors have already added multiple features via pull requests, such as the ability to translate system messages into one of six supported languages or custom messages provided by the user. The active community provides long-term sustainability both for the surveydown project itself and for the research projects that it serves.

## Reproducibility from code

Reproducibility from code is a core advantage of the surveydown platform. The markdown-based approach to survey design is a fundamental change in thinking about how surveys can be created. Rather than using a GUI or spreadsheet interface, surveydown uses plain text files to define all survey content, enabling full reproducibility by default and easy integration with common development tools like Git for version control. By enabling a reproducible workflow for survey construction, surveys can be more easily evaluated by collaborators and reviewers, especially after data collection. For example, the entire survey instrument used in a study can be fully reproduced and experienced by other experts during a peer review process without needing proprietary software, enabling a level of transparency that is difficult or impossible to achieve with other platforms.

Alternative survey platforms do support different forms of reproducibility, but often in limited ways. For example, users of the proprietary software Qualtrics [4] can export a **.qsf** file to share survey designs with other Qualtrics users to reproduce their surveys. However, this only enables reproducibility for users who have a Qualtrics subscription, which limits accessibility and interoperability with version control tools like Git. In contrast, by using plain text files, surveydown surveys can be easily read and directly edited without the need for proprietary software to interpret the files.

Beyond reproducibility, using plain text to define survey content has several other advantages. For example, the survey itself serves as its own documentation since code comments can be used to explain design decisions, which improves long-term maintainability. In comparison, a GUI-based application has limited ability to leave a trail of comments or suggestions about changes. In addition, surveys made using plain text can benefit from using Large

Language Models (LLMs) such as ChatGPT [5] for survey design. The **survey.qmd** and **app.R** files for a survey can be provided survey to an LLM to make revisions and improvements with simple prompts. Likewise, if a user wants to implement a more complex feature than is natively supported, they can use an LLM to help solve how to implement it in Shiny. As AI tools continue to evolve, we expect AI integration with surveydown to become even more significant.

### Programmable interactivity via shiny

Perhaps the most powerful feature of surveydown is its integration with the Shiny web framework, which enables real-time code execution during survey administration. Shiny's reactive programming framework vastly increases the capabilities of surveydown.

A common use case is *conditional control logic*, such as conditionally displaying questions and conditionally navigating to desired pages. For example, consider a multiple choice question where a respondent can select an "other" option that, if chosen, will trigger a second question to display allowing the user to specify the "other" field. This type of control to conditionally display questions is achieved using the `sd_show_if()` function in the **app.R** file, where survey designers can specify any number of conditions that, if true, will display a target question. Likewise, a designer can also conditionally skip a respondent forward to a specified page if a condition is true using the `sd_skip_forward()` function. These functions rely on Shiny's reactive programming framework where logic behavior changes depending on the actions taken by the survey respondent.

Another use case is to reactively change a question label or other text in the survey based on users' previous choices. For example, consider a question asking whether the respondent prefers *dogs* or *cats*; a natural follow-up question might be whether the respondent is a dog or cat owner. Using reactivity, the text of the second question can be dynamically updated (e.g, "do you own a *dog*" versus "do you own a *cat*") depending on what they chose on the first question. We call these "reactive questions," which are defined in the **app.R** file and called in the **survey.qmd** file using `sd_output()`.

Finally, the Shiny framework enables a wide variety of randomization options in how it handles sessions. Respondents can be assigned random values that are held constant for each user or not depending on the survey designer's objective, providing a high degree of flexibility in randomized survey designs.

Because the R Shiny framework is relatively mature, users can also take advantage of all of the existing html widgets developed to create custom questions beyond those already supported. One example of a custom question is an interactive map question using the popular `leaflet` package for creating interactive maps [6]. Users can write R code to define the map widget in the `server()` then pass it as the `output` argument in the `sd_question_custom()` function. Fig 4 below shows a screenshot of an example survey where users are asked to select the state they live in from the map.

### Comparison with existing platforms

In this section we compare surveydown with other popular survey platforms along six categories:

- **User interface**: The interface used by survey designers (not participants).
- **Cost**: Whether the platform is free, paid, or has both free and paid tiers.
- **Reproducibility**: Whether a survey can be fully reproduced from source files.
- **Open-source**: Whether the platform's source code is freely available.

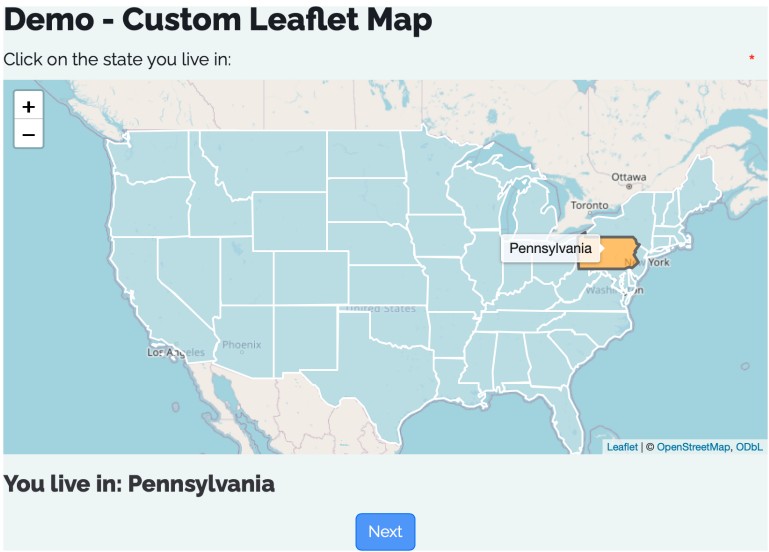

**Fig 4. Screenshot of a custom question using the `leaflet` package to display an interactive map.**

- **Data control**: How much control users have over survey data storage and access.
- **Programmable**: The ability to embed and execute custom code during survey runtime, enabling programmatic control over content display, data processing, and user interactions.

Table 1 compares 14 platforms across these dimensions. For the last four features, we label the feature as "Yes", "No", or "Partially", in which case "Partially" means the platform has some limited capabilities for the feature. For **Reproducibility**, we label a platform as "Partial" if surveys cannot be freely reproduced without proprietary software. For example, while Qualtrics surveys can be reproduced using .qsf files, only Qualtrics subscribers can use them. For **Data Control**, we label a service as "No" if users can only obtain access to the response data through a proprietary service, and "Partial" if the service offers the capability of storing

**Table 1. Comparison of features for select survey platforms.**

| Platform | User Interface | Cost | Reproducible | Open Source | Data Control | Programmable |
|---|---|---|---|---|---|---|
| Google Forms | GUI | Free | □ | □ | □ | □ |
| REDCap | GUI | Free/Paid | ☑ | □ | ■ | ■ |
| Qualtrics | GUI | Paid | ☑ | □ | □ | ■ |
| Sawtooth | GUI | Paid | ☑ | □ | ☑ | ■ |
| CASIC Builder | GUI | Paid | □ | □ | ☑ | □ |
| SurveyCTO | GUI | Paid | ☑ | □ | ☑ | ■ |
| QDS | GUI | Paid | □ | □ | □ | □ |
| LimeSurvey | GUI | Free/Paid | ☑ | ■ | □ | ■ |
| Open Data Kit | XLSForms | Free/Paid | ☑ | ■ | ☑ | ■ |
| oTree | GUI, Python | Free/Paid | ■ | ■ | ■ | ■ |
| SurveyJS | GUI, JavaScript | Free/Paid | ☑ | ■ | ■ | ■ |
| formr | XLSForms, markdown, R | Free | ☑ | ■ | ☑ | ■ |
| shinysurveys | R, CSV | Free | ■ | ■ | ■ | ☑ |
| surveydown | markdown (Quarto), R | Free | ■ | ■ | ■ | ■ |

Legend: ■ = Yes, ☑ = Partially, □ = No.

the data on a private server, which might require a customized or more complex set of steps compared to the service storing the data.

Consider Google Forms [7], a well-known free platform with an intuitive interface for creating simple surveys. While easy to learn, it lacks reproducibility since designs cannot be captured in source files. It is not open-source, provides limited data control with data stored exclusively in Google Sheets, and offers no programmable features. In contrast, the survey-down platform distinguishes itself through its integration with the R ecosystem and Quarto publishing system. It excels in reproducibility through its markdown-based approach, provides full data control, and offers exceptional programmable features through the Shiny framework.

Most platforms in our comparison rely on graphical interfaces or spreadsheet structures (XLSForms) to define survey content, which generally limits reproducibility. Some frameworks like SurveyJS [8] and oTree [9] offer better reproducibility by storing designs as structured data files. Approximately half of the surveyed platforms are open-source, with varying degrees of cost, data control options, and programmable features. Notable open-source alternatives to surveydown include formr [10], which also integrates with R but requires a complex server setup for self-hosting; LimeSurvey [11], which offers extensive features as a GUI-based platform, and Open Data Kit [12], which excels in field-based mobile data collection but creates a disconnect between survey design and analysis environments.

Finally, within the R ecosystem specifically, several approaches have emerged that also leverage Shiny for survey implementation. Kaufman (2020) highlighted the potential of R-based survey tools with Shiny using a series of examples, but did not provide a comprehensive package [13]. A close alternative to surveydown is shinysurveys package, by Trattner and D'Agostino McGowan (2021), which offers a more formalized implementation comparable to Google Forms, and light programmability support with R code. The approach provides reproducibility but with relatively simple functionality limited to basic survey designs that rely on predefined functions and structures, offering less flexibility for complex survey designs and custom interactive elements [14].

Given the flexibility of the Shiny web framework, surveydown can also serve as a free and open-source alternative to existing proprietary platforms for more specialized purposes. For example, the Poll Maker by Mentimeter [15] is a popular proprietary platform for creating interactive live polls and quizzes, where respondents see the live survey results in real time. A similar live polling capability can be achieved with surveydown using the `sd_get_data()` function, which gets the latest response data and refreshes according to a specified time interval, which can then be used to display summary results to respondents. A live-polling template is available at https://surveydown.org/templates/live_polling.

Finally, it is important to note security considerations for data collection tools like survey-down. Given surveydown's disaggregated design, three separate components require security considerations: the surveydown application code, the app hosting service, and the data storage service. For the surveydown application code, we have followed best practices in how survey response data is internally handled, such as using SQL injection prevention strategies and ensuring that users store their database credentials as a .env file to avoid accidental exposure. We also adopted an architecture where all content in the survey is served entirely from the shiny server, preventing respondents from being able to see content in the source code of other pages before getting there from the survey navigation. While the package does not yet have a security compliance certificate for the application code, this is a longer-term goal. For the app hosting service, users can choose from different providers, each of which offer different security measures. For example, while shinyapps.io is a free service, it is not HIPAA compliant. Alternatives such as Heroku or Hugging Face may offer other security measures, and

users are encouraged to review their security needs before choosing a hosting service. Finally, for data storage, we suggest Supabase as a free, open-source, and convenient to use platform that has TLS encryption among other security features, including multi-factor authentication (MFA) and being SOC 2 and HIPAA compliant.

## Discussion and conclusion

The surveydown platform represents a significant step forward in survey methodology by bringing the principles of reproducible research to survey design and implementation. By combining the expressiveness of markdown, the computational power of R, and the interactivity of Shiny, surveydown enables researchers to create sophisticated survey instruments that are fully documented, version-controlled, and integrated with data analysis workflows.

One of the primary contributions of surveydown is the idea of achieving full reproducibility through code. By defining surveys in plain text (markdown and R code), surveydown enables complete reproducibility and version control of survey instruments, supporting transparent research practices and long-term preservation of survey instruments. The platform also offers programmable interactivity by leveraging the Shiny web framework, enabling real-time code execution during survey administration. Another significant contribution is its open and disaggregated architecture that separates survey design, deployment, and data storage, giving researchers control over each component. Finally, as a free and open-source platform, surveydown removes financial barriers to sophisticated survey research tools while providing extensive customization options and benefiting from a growing community of contributors.

The platform does have limitations. Since the platform requires some minimal knowledge of markdown and R, it creates a higher entry barrier compared to typical GUI-based platforms, potentially limiting adoption among researchers without coding experience. Additionally, while the disaggregated architecture provides flexibility, it also requires users to handle deployment and database configuration, which may be challenging for some users, though careful documentation and tutorials helps ease these barriers. From a functionality perspective, although surveydown can currently handle complex survey designs, it does not yet match all specialized features of mature platforms, especially with respect to participant recruitment and tracking as some platforms do (e.g., Qualtrics). Finally, as with any web-based survey platform, performance under high concurrent loads depends on the hosting service chosen by the user. While free hosting options like shinyapps.io exist, users may need to pay for greater hosting server performance.

Looking forward, surveydown has several promising future directions. A graphical interface that creates the **survey.qmd** and **app.R** files for a surveydown survey would lower the entry barrier for users with limited coding experience while maintaining a reproducible workflow. This approach would bridge the gap between ease-of-use and reproducibility, potentially broadening the platform's appeal. We are in the process of building this tool as a companion package called sdstudio (https://github.com/surveydown-dev/sdstudio), which will serve as a comprehensive studio for building, previewing, and managing surveys using a GUI while maintaining full reproducibility. In addition, building a comprehensive library of templates for common survey designs would accelerate implementation for new users and promote best practices in survey design. We have already begun this with a series of existing templates available on the main documentation website at https://surveydown.org. Also, while the current implementation is responsive, developing specialized mobile question types and

layouts would improve the experience for respondents on mobile devices, which represent an increasing share of survey participants.

As survey research continues to evolve, platforms that emphasize transparency, reproducibility, and programmatic flexibility will become increasingly important. The open-source nature of surveydown ensures that it can grow alongside changing methodological requirements and technological capabilities, driven by the needs of the research community it serves. By reimagining survey design through code, surveydown not only addresses practical limitations in existing platforms but also aligns survey methodology with broader trends toward computational reproducibility in scientific research. This approach has the potential to enhance the rigor, transparency, and long-term value of survey-based research across disciplines.

## Acknowledgments

We would like to thank the two reviewers for their helpful suggestions on this manuscript. We also gratefully acknowledge the early adopters of surveydown—Reed Benkendorf, Jeffrey Girard, Will King, Stefan Munnes, Robert Kubinec, and Christian Willig—for their valuable feedback and support.

## Author contributions

**Conceptualization:** John Paul Helveston.

**Data curation:** John Paul Helveston.

**Formal analysis:** Pingfan Hu, Bogdan Bunea.

**Funding acquisition:** John Paul Helveston.

**Investigation:** Pingfan Hu.

**Methodology:** John Paul Helveston.

**Project administration:** John Paul Helveston.

**Resources:** Pingfan Hu, John Paul Helveston.

**Software:** Pingfan Hu, Bogdan Bunea, John Paul Helveston.

**Supervision:** John Paul Helveston.

**Validation:** Pingfan Hu.

**Visualization:** John Paul Helveston.

**Writing – original draft:** Pingfan Hu, John Paul Helveston.

**Writing – review & editing:** Pingfan Hu, Bogdan Bunea, John Paul Helveston.

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
