## [Decision Letter · Decision Letter 0]

2 Jul 2025

PONE-D-25-20441surveydown: An Open-Source, Markdown-Based Platform for Programmable and Reproducible SurveysPLOS ONE

Dear Dr. Helveston,

Thank you for submitting your manuscript to PLOS ONE. After careful consideration, we feel that it has merit but does not fully meet PLOS ONE’s publication criteria as it currently stands. Therefore, we invite you to submit a revised version of the manuscript that addresses the points raised during the review process.

 I agree with a reviewer about the need for further revisions. Please submit your revised manuscript by Aug 16 2025 11:59PM. If you will need more time than this to complete your revisions, please reply to this message or contact the journal office at plosone@plos.org. Please include the following items when submitting your revised manuscript:

We look forward to receiving your revised manuscript.

Kind regards,

Diego A. Forero, MD; PhD

Academic Editor

PLOS ONE

Journal Requirements:

[This work was partially supported by a grant from the Alfred P. Sloan Foundation (https://sloan.org/), Grant Number G-2023-20976 awarded to PI John Paul Helveston. The funders did not play a role in the study design, data collection and analysis, decision to publish, or preparation of the manuscript.].

[This research was supported by the Alfred P. Sloan Foundation (https://sloan.org/) under Grant No. G-2023-20976 awarded to PI John Paul Helveston.]

[This work was partially supported by a grant from the Alfred P. Sloan Foundation (https://sloan.org/), Grant Number G-2023-20976 awarded to PI John Paul Helveston. The funders did not play a role in the study design, data collection and analysis, decision to publish, or preparation of the manuscript.]

5. We note that Figure 4 in your submission contains a map image which may be copyrighted. All PLOS content is published under the Creative Commons Attribution License (CC BY 4.0), which means that the manuscript, images, and Supporting Information files will be freely available online, and any third party is permitted to access, download, copy, distribute, and use these materials in any way, even commercially, with proper attribution. For these reasons, we cannot publish previously copyrighted maps or satellite images created using proprietary data, such as Google software (Google Maps, Street View, and Earth). For more information, see our copyright guidelines: http://journals.plos.org/plosone/s/licenses-and-copyright.

1. You may seek permission from the original copyright holder of Figure 4 to publish the content specifically under the CC BY 4.0 license. 

Reviewers' comments:

Reviewer's Responses to Questions

**Comments to the Author**

1. Is the manuscript technically sound, and do the data support the conclusions?

Reviewer #1: Yes

Reviewer #2: Yes

2. Has the statistical analysis been performed appropriately and rigorously? 

Reviewer #1: N/A

Reviewer #2: N/A

3. Have the authors made all data underlying the findings in their manuscript fully available?

Reviewer #1: Yes

Reviewer #2: Yes

4. Is the manuscript presented in an intelligible fashion and written in standard English?

Reviewer #1: Yes

Reviewer #2: Yes

5. Review Comments to the Author

Reviewer #1: The authors present an R package for building and managing surveys using a markdown based system. The distinguishing features of the package over existing methods is the comprehensive nature of the package, including git version control and real-time code execution through shiny and control over the implementation details without needing web development skills. The package generates the web application completely using R which is very useful for data scientists who are often unfamiliar with writing html and using javascript frameworks.

I found the design to be very well thought out and logical, using existing formats and well established standard packages Quarto, Shiny and Postgres. There is a useful and extensive discussion of the existing packages and the differences which was clear and compelling, especially with the differences over the most obvious alternative - Google Forms. I found that the focus on making it reproducible through being able to specify everything through code is extremely useful for those trying to tie surveys with other software packages. The authors also state that the package could have a GUI front-end if necessary which also enhances its utility. The availability of templates for surveys would also encourage adoption.

Reviewer #2: I think surveydown can be a great tool for easy survey programming. I just have a few outstanding issues about the software and about the manuscript.

Concerning your app.R file: suppose that I make multiple surveys with surveymarkdown and run sd_ui(). How does R know which .qmd file I want to run? In other words, what connects a particular app.R document to a particular survey.qmd document? Is it just that they need to be in the same directory/project? And if so, can each directory only accommodate one survey.qmd document? (I would not perceive this as a shortcoming of the software, as one can always just make more directories; I’d just like it spelled out for readers.)

An issue that should not be left to be figured out later is data security. Platforms like Qualtrics use TLS encryption for data transmission. You envision that surveydown platforms will deploy on using either personal or third-party hosting services, who then pass responses from a deployed Shiny app back to the survey creator’s PostgreSQL database. Does the encryption and security with which that information is passed vary by hosting provider? If so, can you shed some light on which hosting platforms offer that kind of encryption, and/or which ones may not be appropriate for surveys that aim to collect sensitive information?

I see that you explicitly recommend supabase for SQL storage. Can you clarify the security features on supabase?

Make sure to update the GitHub figures (stars, issues, discussions) for the next iteration of the manuscript.

The section on reproducibility of code doesn’t give enough credit to the inter-interpretability of other survey platforms. My survey platform of choice is typically Qualtrics, and other researchers who use Qualtrics can pretty easily figure out the structure of eachother’s Qualtrics surveys by reuploading .qsf files into Qualtrics and reading the survey structure in the GUI. This section could be improved by just clarifying that with plain text structuring, this sort of inter-interpretability arises *even without access to proprietary software* (like Qualtrics).

In Section 3.3, you talk about using conditional logic via Shiny. It’s clear that a page can be displayed only if a logical condition is true, but how could a user reverse the logical condition being true to view older pages and change answers? In other words, how do you implement a ‘back button’ in surveydown? This is a feature many social scientists consider necessary in their surveys, especially if they arise from complex experiments.

Relatedly, if a ‘back button’ is possible in surveydown, how do survey designers control logical conditions that hold randomly? Survey platforms like Qualtrics will fix a random seed for each individual participant, preventing someone from changing their randomly-assigned treatment status by using the back button and receiving another random draw. How can survey users implement similar protocols in surveydown?

Prose error on line 292: “We call these questions ‘reactive question’”

I think the live-polling feature of surveydown could make it a useful replacement for proprietary live-polling platforms such as Menti. Do you agree? If so, I think this is a use case worth highlighting, as universities currently shell out considerable funds to give teachers collective access to such GUI-based software.

Table 1 is useful but the categorization is rather vague. Rather than binning platforms into each category via ‘yes’, ‘partially’, and ‘no’ labels, the paper would benefit from these labels being expanded out, explaining what you mean in each. This also gives users a clearer demonstration of the differences between software they’re already using and a software they might want to switch to (say, surveydown).

I don’t think it’s fair to single out LimeSurvey for requiring users to know MySQL for customization when your software requires PostgreSQL for data management.

In the conclusion, what do you mean by ‘built-in panel management’ and ‘advanced quota management’?

6. PLOS authors have the option to publish the peer review history of their article (what does this mean?). If published, this will include your full peer review and any attached files.

Reviewer #1: **Yes: **Ling-Hong Hung

Reviewer #2: **Yes: **Jack Fitzgerald

---

## [Author Response · Author response to Decision Letter 1]

4 Jul 2025

Please see our detailed response to the reviewers included as a separate pdf.

---

## [Decision Letter · Decision Letter 1]

30 Jul 2025

PONE-D-25-20441R1surveydown: An Open-Source, Markdown-Based Platform for Programmable and Reproducible SurveysPLOS ONE

Dear Dr. Helveston,

Thank you for submitting your manuscript to PLOS ONE. After careful consideration, we feel that it has merit but does not fully meet PLOS ONE’s publication criteria as it currently stands. Therefore, we invite you to submit a revised version of the manuscript that addresses the points raised during the review process.

We look forward to receiving your revised manuscript.

Kind regards,

Diego A. Forero, MD; PhD

Academic Editor

PLOS ONE

Journal Requirements:

Additional Editor Comments:

I agree with a reviewer about the need for a further minor revision.

Reviewers' comments:

Reviewer's Responses to Questions

**Comments to the Author**

1. If the authors have adequately addressed your comments raised in a previous round of review and you feel that this manuscript is now acceptable for publication, you may indicate that here to bypass the “Comments to the Author” section, enter your conflict of interest statement in the “Confidential to Editor” section, and submit your "Accept" recommendation.

Reviewer #1: (No Response)

Reviewer #2: (No Response)

2. Is the manuscript technically sound, and do the data support the conclusions?

Reviewer #1: Yes

Reviewer #2: Yes

3. Has the statistical analysis been performed appropriately and rigorously? 

Reviewer #1: N/A

Reviewer #2: N/A

4. Have the authors made all data underlying the findings in their manuscript fully available?

Reviewer #1: Yes

Reviewer #2: (No Response)

5. Is the manuscript presented in an intelligible fashion and written in standard English?

Reviewer #1: Yes

Reviewer #2: (No Response)

6. Review Comments to the Author

Reviewer #1: I found the revisions to be useful and the comments helpful in understanding the scope and rationale for their software.

Reviewer #2: This looks good to go on two conditions. (1) Parse for spelling in the new additions (e.g., HIPPA vs. HIPAA). (2) expand your definition of 'programmable' in Table 1 to include a 'partial' category to include lightly-programmable software like shinysurveys, and provide a good definition of partial programmability for Table 1.

7. PLOS authors have the option to publish the peer review history of their article (what does this mean?). If published, this will include your full peer review and any attached files.

Reviewer #1: **Yes: **Ling-Hong Hung

Reviewer #2: **Yes: **Jack Fitzgerald

---

## [Author Response · Author response to Decision Letter 2]

5 Aug 2025

Please see the attached response to reviewers as a pdf

---

## [Decision Letter · Decision Letter 2]

11 Aug 2025

surveydown: An Open-Source, Markdown-Based Platform for Programmable and Reproducible Surveys

PONE-D-25-20441R2

Dear Dr. Helveston,

We’re pleased to inform you that your manuscript has been judged scientifically suitable for publication and will be formally accepted for publication once it meets all outstanding technical requirements.

Kind regards,

Diego A. Forero, MD; PhD

Academic Editor

PLOS ONE

Additional Editor Comments (optional):

Reviewers' comments:

Reviewer's Responses to Questions

**Comments to the Author**

1. If the authors have adequately addressed your comments raised in a previous round of review and you feel that this manuscript is now acceptable for publication, you may indicate that here to bypass the “Comments to the Author” section, enter your conflict of interest statement in the “Confidential to Editor” section, and submit your "Accept" recommendation.

Reviewer #2: All comments have been addressed

2. Is the manuscript technically sound, and do the data support the conclusions?

Reviewer #2: Yes

3. Has the statistical analysis been performed appropriately and rigorously? 

Reviewer #2: N/A

4. Have the authors made all data underlying the findings in their manuscript fully available?

Reviewer #2: Yes

5. Is the manuscript presented in an intelligible fashion and written in standard English?

Reviewer #2: Yes

6. Review Comments to the Author

Reviewer #2: I believe this addresses all concerns I have. I wish you the best of luck with further developing the software.

7. PLOS authors have the option to publish the peer review history of their article (what does this mean?). If published, this will include your full peer review and any attached files.

Reviewer #2: **Yes: **Jack Fitzgerald

---

## [Editor Report · Acceptance letter]

PONE-D-25-20441R2

PLOS ONE

Dear Dr. Helveston,

I'm pleased to inform you that your manuscript has been deemed suitable for publication in PLOS ONE. Congratulations! Your manuscript is now being handed over to our production team.

Kind regards,

on behalf of

Dr. Diego A. Forero

Academic Editor

PLOS ONE